# Field Validation of Commercially Available Food Retailer Data in the Netherlands

**DOI:** 10.3390/ijerph17061946

**Published:** 2020-03-16

**Authors:** Cesare Canalia, Maria Gabriela M. Pinho, Jeroen Lakerveld, Joreintje D. Mackenbach

**Affiliations:** 1Department of Epidemiology and Biostatistics, Amsterdam UMC, Vrije Universiteit Amsterdam, 1117 de Boelelaan, Amsterdam, The Netherlands; cesare.canalia@gmail.com (C.C.); m.matiasdepinho@amsterdamumc.nl (M.G.M.P.); j.lakerveld@amsterdamumc.nl (J.L.); 2Upstream Team, Amsterdam UMC, 1117 de Boelelaan, The Netherlands; 3Julius Center for Health Sciences and Primary Care, University Medical Center Utrecht, 3584 CX Utrecht, The Netherlands; 4Faculty of Geosciences, Department of Human Geography and Spatial Planning, Utrecht University, 3508 TC Utrecht, The Netherlands

**Keywords:** validity, retail food environment, foodscape, street audit, ground-truthing

## Abstract

The aim of this study was to validate a Dutch commercial dataset containing information on the types and locations of food retailers against field audit data. Field validation of a commercial dataset (“Locatus”) was conducted in February 2019. Data on the location and classification of food retailers were collected through field audits in 152 streets from four urban and four rural neighborhoods in the Netherlands. The classification of food retailers included eight types of grocery stores (e.g., supermarkets, bakeries) and four types of food outlets (e.g., cafés, take away restaurants). The commercial dataset in the studied area listed 322 food retailers, whereas the field audit counted 315 food retailers. Overall, the commercially available data showed “good” to “excellent” agreement statistics (>0.71) with field audit data for all three levels of analysis (i.e., location, classification and both combined) and across urban as well as rural areas. The commercial dataset under study provided an accurate description of the measured food environment. Therefore, policymakers and researchers should feel confident in using this commercial dataset as a source of secondary data.

## 1. Introduction

The food environment plays an important role in shaping dietary habits, and consequently people’s health [1]. The food environment is defined as “the multiplicity of sites where food is displayed for purchase, and where it may also be consumed” [2], and may both enhance and inhibit healthy dietary behaviors through ubiquitous opportunities to purchase and consume large varieties of healthy and unhealthy foods [3]. Given the potential opportunities to improve population diets and health outcomes, there is increasing interest in studies on the influence of the food environment on dietary behaviors [4,5].

The majority of the studies that have examined the link between food environment and health-related outcomes have focused on the geographical availability or accessibility of food retailers in relation to where people live and/or work [5]. From a public health perspective, the type of food retailers that exist near residential areas and schools may be especially relevant to the youth. For example, a greater availability and proximity to unhealthy food retailers has been associated with obesity and other diet-related chronic diseases, such as type 2 diabetes and cardiovascular disease [6,7,8,9,10,11,12]. Likewise, greater distance to fast food/convenience shops and proximity to retailers selling healthier foods has been shown to be associated with healthier dietary habits and better health statuses [9,13,14,15]. However, the current body of research in this field is inconsistent [6,16,17,18,19,20,21,22,23,24,25,26,27,28], with many studies showing null or counterintuitive results.

A recent literature review has suggested various possible causes for the lack of consistent results found in food environment research. These potential causes include the methodological procedures employed, as well as the quality of the datasets used [29], which is further considered in the current study. Data on the geographical location of food retailers can be obtained from various sources. Primary data sources are gathered directly by research groups as systematic observation and notation of surrounding features, usually while walking through a specific area. Primary data sources are also known as field audits or ground-truthing, and are described as the “gold standard” in the field of food environment research [30,31]. Secondary data sources are collected by an external party to the research group and usually have purposes beyond health research objectives [32]. Examples of this include commercial retail data, yellow pages, and governmental repositories.

Even though secondary data sources are readily available and relatively accessible [29], they have some of the following limitations: they may provide incomplete information about food retailers; they may use a different classification of food retailers than those needed for health research purposes; they are often not freely available; and they may not be updated regularly [33,34,35]. Conversely, despite primary data sources being described as the “gold standard”, they are not without limitations. For example, since extensive fieldwork is required to obtain them, collecting primary data is costly and time-consuming [36]. Hence, studies exploring the relationships between food environment and health outcome have largely relied on secondary data sources [29], raising questions regarding their validity.

Previous studies have performed validation analyses of secondary data sources in various settings, countries, and populations [37,38], but this is still not a common practice in food environment research. Moreover, since food environment research is increasing, much work still remains to be done in order to identify which secondary food retailer data sources are most appropriate. To the best of our knowledge, no empirical evidence concerning the validity of secondary data sources exists in the Netherlands. Therefore, in order to help policymakers and researchers make informed decisions about which data sources to use, we aimed to test the validity of a Dutch commercial dataset containing information on geographical locations and types of food retailers against field audit data.

## 2. Materials and Methods

Since this validation study did not involve individual level data, it was exempt from approval by the Medical Ethics Committee of the Vrije Universiteit Medical Center in Amsterdam, the Netherlands.

### 2.1. Study Areas

The Netherlands is geographically divided into four regions [39]. The North Netherlands comprises the provinces Groningen, Friesland, and Drenthe; the East Netherlands comprises Overijssel, Gelderland, and Flevoland; the West Netherlands comprises Utrecht, North Holland, South Holland, and Zeeland; and the South Netherlands comprises North Brabant and Limburg.

To obtain an accurate representation of the Dutch food environment, we selected eight neighborhoods in total, two in each of the four above-mentioned regions, four being in an urban area, and four in a rural area. Both urban and rural areas were selected, as the validity of food environment data may differ across urbanization levels [38]. In this regard, urbanization level information of Dutch neighborhoods was accessed via the Dutch Centraal Bureau voor de Statistiek (CBS) in February 2019 [40]. CBS distinguishes different levels of urbanization based on the number of addresses per km^2^ [41]: very strongly urbanized areas (environmental address density of 2500 or more), strongly urbanized areas (environmental address density from 1500 to 2500), moderately urbanized areas (environmental address density from 1000 to 1500), low urbanized areas (environmental address density from 500 to 1000), and non-urban areas (environmental address density of less than 500) [40]. For the sake of this study, we considered very strongly and strongly urbanized areas as “urban” areas, and low and non-urbanized areas as “rural” areas.

Since the field audit was conducted on foot by one auditor, and due to research time constraints, the sampling was based on relatively small and similar sized areas that were accessible by public transport. This convenience sampling process resulted in selection of the following eight neighborhoods: Binnenstad-Noord (urban; Groningen), Oosterhaar (rural; Haren), Binnenstad (urban; Oldenzaal), Neede (rural; Berkelland), Binnenstad-Centrum (urban; ‘s-Hertogenbosch), Vliedberg (rural; Heusden), Anjeliersbuurt Noord/Zuid and Driehoekbuurt (urban; Amsterdam), and Ouderkerk aan de Amstel (rural; Ouder-Amstel).

In each of the predefined eight neighborhoods, 19 streets were selected for comparison with field audits, resulting in a total of 152 streets. The selection of streets was performed by a researcher that did not conduct the field audit. Within each selected neighborhood, streets that contained at least one food retailer were randomly selected from the Locatus dataset. Streets with no food retailer present were not included in the Locatus dataset. However, it is important that the correspondence between absence of food retailers according to both data sources could also be assessed. Therefore, streets with no food retailer present were selected from Google Maps. Streets were selected in such a way that 91 of these streets had at least one food retailer present, while the other 61 streets did not contain any food retailer. If a selected street crossed a neighborhood boundary, the given street was audited only until the limits of the selected neighborhood were reached.

### 2.2. Data Sources

Data on food retailers were obtained from Locatus [42], a Dutch company that collects information on different types of retail outlets for commercial purposes. Locatus covers all of the Netherlands and is widely used among retailers, policymakers, and researchers. Locatus collects information on location, type, size, and opening times of all retailers through systemic area scans, which are conducted by employees of Locatus via field audits. Food outlets in shopping areas are audited every year, while food outlets in scattered shopping areas are audited every two to three years, as the presence of retailers located in these areas tends to be more stable. When an audit takes place, all food retailers present in that neighborhood are assessed. In addition, office worker employees of Locatus conduct surveys and telephone interviews to receive updates about retailers, which makes it likely that changes in the food environments of both shopping and scattered shopping areas are noted within a year. In order to minimize the mismatch between the available data and field audit data in this study, the latest available data dating from July 2018 were used.

The field audit was conducted between 22 February and 2 March 2019. For the collection of the field audit data, the first author was instructed to scan the study area on foot in order to systematically observe and note the surrounding features. A standardized protocol was developed among the involved researchers to guide the field audit. The field audit proceeded as follows: (1) both sides of all selected streets were audited on foot; (2) the name, type, and location of each food retailer were recorded on a digital checklist; (3) establishments were classified based on external clues (e.g., names and signs); (4) in case of doubt, the auditor was instructed to enter the food retailer, consult the menu to identify main meals or products sold, or check the opening hours, assuming that, for instance, a full-service restaurant usually opens after 17:00. Establishments that had a sign indicating permanent closure or that appeared to be permanently closed were not considered to be present in the field. Importantly, the auditor was blinded to the information provided by the commercial dataset (only the street name was known), in order to prevent bias.

### 2.3. Field Audit Classification of Food Retailers

As shown in Table 1, two main categories (grocery stores and food outlets) were constructed and further divided into 12 subcategories of food retailers based on the food retailer classifications developed by Clary and Kestens [43] and Locatus’s definitions. These classifications were constructed prior to the field audit process.

Specifically, the grocery stores were classified and defined as follows:Supermarkets: large food store chains selling a wide range of fresh, packaged, and frozen food products;Local product shops: independent food stores selling a wide range of local and/or ethnic food products from EU and non-EU countries;Fruit and vegetable stores: food stores selling mostly fruits, vegetables, and nuts;Bakeries: food stores selling bread and other baked products;Animal product stores: food stores selling mainly animal products, such as meat, dairy, or fish;Natural product stores: food stores selling mostly superfoods, food supplements, homeopathic products, herbs, or coffee/tea;Convenience stores: food stores selling a limited range of fresh and healthy food products, and primarily offering snack foods;Confectionery stores: food stores specialized in selling a wide assortment of sweet products, including pastries, chocolates, candies, and ice-cream.

Similarly, the food outlets were given the following classifications based on the following characteristics:Restaurants: chain or independent restaurants with à la carte menu or buffet, offering ready-to-eat foods with table service or the possibility to sit at a table;Fast food restaurants: chain or independent restaurants with counter service only and limited seating options, selling mainly cheap ready-to-eat high energy density foods served a few minutes after ordering;Take away restaurants: chain or independent restaurants where ready-to-eat food is delivered or picked up with no or limited seating options;Cafés: chain or independent retailers offering alcoholic/non-alcoholic beverages and serving ready-to-eat sweet/salty snacks and meals, with the possibility to sit in and/or take away.

Although there are many retailers that have dual purposes (e.g., a fast food restaurant that also delivers meals), retailers were classified on the basis of their main purpose. Establishments whose business purpose was to sell beverages only, such as bars and liquor stores, were not considered to be food retailers in the present study. In addition, street vendors, such as food trucks and market stalls, were excluded from the classification system due to their itinerant nature.

### 2.4. Statistical Analyses

First, matching food retailers needed to be identified. In order do so, the Locatus and the audit datasets were merged into one file, and streets listed in the commercial dataset were compared to the field audit data. Three different matching levels were considered: “location”, “classification”, and both combined. For instance, a match between two food retailers was established when they were either present in the same location (i.e., according to street name and house numbering), had the same classification, or shared both a location and classification. In other words, food retailers listed in the commercial dataset were defined as true positives (TPs) if they were equally classified or found to be in the exact same location by the field audit. Non-matches were interpreted as false positives (FPs) if food retailers were listed in the commercial dataset but did not match with the field audit data, or vice versa (i.e., false negatives (FNs)). The number of “empty” streets in which food retailers were found neither by the commercial dataset nor by the field audit were referred as true negatives (TNs). Importantly, if there were spelling differences in the business name, discrepancies in the street name because a food retailer was located at a street junction, or errors in the house numbering, the retailer was still considered a match. This approach is known as “relaxed” matching criteria, and has been described as the most appropriate method when investigating the validity of a dataset, since the specific retailer name or exact address are of minor importance [43,44].

Next, agreement statistics such as sensitivity, specificity, positive predictive value (PPV), Cohen’s kappa, and concordance were estimated at the street level (see Table 2). Sensitivity reflected the ability of a data source to correctly capture food retailers that were actually present in the field, and was determined by the proportion of food retailers present in the field that were listed in the commercial dataset. Specificity was determined by the proportion of true negatives (i.e., empty streets) that were correctly classified as not presenting food retailers. PPV reflects the proportion of listed food retailers that were also present in the field (food retailers observed in the field that were not listed in the commercial dataset were not considered). Cohen’s kappa measured the agreement between field audit and Locatus data, taking into account agreements that occurred by chance. Concordance assessed the proportion of food retailers listed both in the commercial dataset and present in the field among all the food retailers present.

Finally, for each of the three matching levels (location, classification, and both combined), agreement statistics were calculated for all food retailers combined, all food retailers combined but stratified by urban and rural areas, and separate food retailer subcategories. The level of agreement was interpreted using the following criteria: <0.30 was considered “poor”, 0.31–0.50 was “fair”, 0.51–0.70 was “moderate”, from 0.71–0.90 was “good”, and >0.90 was “excellent” [45]. The dataset did not contain missing data. Statistical analyses were performed using SPSS version 25.0 (IBM Corp, 2017, Armonk, NY, USA).

## 3. Results

### 3.1. Descriptive Statistics

In the 152 selected streets, the Locatus dataset indicated 322 food retailers to be present, of which 276 were located in urban areas and 46 were located in rural areas. Of the 322 food retailers listed, 5.3% were supermarkets, 0.6% were fruit and vegetable stores, 3.1% were bakeries, 6.8% were animal product stores, 2.5% were natural product stores, 1.9% were convenience stores, 4% were confectionery stores, 42.5% were restaurants, 10.2% were fast food restaurants, 6.8% were take away restaurants, and 16.1% were cafés (no local product shops were listed in the Locatus dataset). Via the field audit, 315 food retailers were identified, of which 265 were located in urban areas and 50 were located in rural areas. Of the 315 food retailers analyzed, 4.8% were supermarkets, 1.9% were local product shops, 1% were fruit and vegetable stores, 2.2% were bakeries, 5.7% were animal product stores, 3.2% were natural product stores, 1% were convenience stores, 2.9% were confectionery stores, 48.3% were restaurants, 9.5% were fast food restaurants, 4.1% were take away restaurants, and 15.6% were cafés (see Table 3). 

Of the 322 food retailers present in the Locatus dataset, 246 matched the food retailers found in the field, while 1 food retailer had a wrong address (1 animal product store). A total of 42 were wrongly classified (1 bakery, 8 animal product stores, 1 natural product stores, 3 convenience stores, 3 restaurants, 4 fast food restaurants, 8 take away restaurants, and 14 cafés). In most instances, there was a “close” mismatch, such as a fast food outlet classified as a take away outlet, or a café being classified as a restaurant. In some instances, a mismatch in store name indicated a replacement of the store/restaurant. There were 33 food outlets not found in the field (2 supermarkets, 2 bakeries, 6 confectionery stores, 13 restaurants, 2 fast food restaurants, 4 take away restaurants, and 4 cafés). In addition, 26 of the 315 food retailers found in the field were not listed in Locatus, of which 1 was a local product shop, 1 was a fruit and vegetable store, 3 were animal product stores, 1 was a natural product store, 1 was a confectionery store, 10 were restaurants, 1 was a take away restaurant, and 8 were cafés.

### 3.2. Agreement Statistics on the “Location” of Food Retailers

Overall, sensitivity of the location of food retailers was “excellent” (0.914), and PPV and concordance were “good” (0.897; 0.827) (see Table 4). Agreement statistics stratified by urbanization levels showed that in urban areas sensitivity was “excellent” (0.921), and PPV and concordance were “good” (0.887; 0.824). In rural areas, sensitivity was “good” (0.880), PPV was “excellent” (0.957), and concordance was “good” (0.846).

Agreement analyses were also conducted for each of the 12 food retailer subcategories. “Good” to “excellent” sensitivity was observed across all subcategories, except for fruit and vegetable stores (0.667). “Good” to “excellent” PPV was detected for all subcategories, except for confectionery stores (0.538). “Good” to “excellent” concordance was detected for all subcategories, except for fruit and vegetable stores (0.666) and confectionery stores (0.500).

### 3.3. Agreement Statistics on the “Classification” of Food Retailers

As shown in Table 5, overall PPV for the classification of food retailers was “good” (0.855). Agreement statistics stratified by urbanization level highlighted that PPV was “good” in both urban and in rural areas (0.849, 0.886).

Agreement analyses were also conducted for each of the 12 food retailer subcategories. “Good” to “excellent” PPV was observed across all subcategories, except for animal product stores (0.636), convenience stores (0.500), and take away restaurants (0.556).

For this analysis, only matching food retailers (retailers that were identified by both the commercial dataset and field audit) were considered. Indeed, retailers that were listed in Locatus but not found in the field, or present in the field but not in Locatus, as well as “empty” streets, could not be considered in this analysis because they had no counterparts to be compared with for their classification. In other words, only true positives and false positives were included, since the field audit could only confirm (true positive) or disconfirm (false positive) the classification given by Locatus. Therefore, the only agreement value that could be estimated was the PPV (= TP/TP + FP).

### 3.4. Agreement Statistics on Both “Location and Classification” of Food Retailers

Overall, sensitivity for both the locations and classification of food retailers was “excellent” (0.996), and PPV was “good” (0.854) (see Table 6). Similarly, agreement statistics stratified by urbanization level showed that in both urban and rural areas, sensitivity was “excellent” (0.995 and 1.000, respectively) and PPV was “good” (0.848 and 0.886, respectively).

Agreement analyses were also conducted for each of the 12 food retailer subcategories. “Excellent” sensitivity was observed across all subcategories. “Good” to “excellent” PPV was measured for all subcategories, except for animal product stores (0.619), convenience stores (0.500), take away restaurants (0.556), and cafés (0.708).

Again, only matching food retailers (retailers that were identified by both the commercial dataset and the field audit) were considered. Specificity and Kappa statistics could not be determined to assess agreement with the field audits, because the number of “true negatives” (i.e., streets with no food retailers) in this analysis was not applicable.

## 4. Discussion

In the present study, we aimed to evaluate the validity of a secondary data source (“Locatus”) containing information on the geographical locations of food retailers against a field audit. Our main results showed that these commercially available data had overall “good” to “excellent” agreement statistics as compared to the field audit data for all three levels of analysis (i.e., location, classification and both combined).

Previous studies have performed validation analysis of secondary data sources [37,38]. In general, commercial retail data and governmental repositories have been reported to have greater validity than other secondary data sources (e.g., yellow pages) [37], with most studies [38] reporting validity scores comparable with the results of the present study. Given the increasing interest in food environment research, transparency about the quality of data sources is of utmost importance. Although conclusions about the validity of secondary data sources may obviously differ due to the characteristics of the specific data under study, some of those differences may be explained by the methodological choices of the researchers and commercial parties. For example, mismatches between the secondary data and field audit data could be related to the temporal difference of the data collection between the two sources (e.g., some shops closed or changed names between the data collection of Locatus and the field audit) and the definition and interpretation of types of food retailers. In turn, insight into these methodological choices may facilitate comparisons and the potential harmonization of food environment datasets across settings and regions.

One methodological choice to be made by researchers pertains matching criteria. For instance, while several studies have used field audits as the “gold standard” [36,38,43,44,46,47,48,49,50] to explore the validity of secondary data sources, only some adopted “relaxed” matching criteria [38,43,44,47]. “Relaxed” matching criteria tolerates mismatches due to discrepancies in business names or slight imprecisions in location [43,44]. However, when considering every single mismatch due to business name and location error (e.g., retailer present on the right street but listed with wrong house number), an underestimation of the validity of a dataset may occur. As such, the choice of the matching criteria may to some extent explain differences in conclusions about the validity of secondary data sources [36,38,46].

Another methodological choice to be made by researchers is the area under study, thereby balancing precision and feasibility of the study. As the validity of food environment data may vary across urbanization levels, we explored possible discrepancies between urban and rural areas. Since the frequency of permanent closure of retailers may be higher in rural areas than urban areas, commercially available data have been described as possibly having greater validity in the latter than the former, since it is not able to capture these changes [51]. Nevertheless, we did not find considerable differences in agreement statistics between urban and rural areas. Few studies have compared the validity of secondary data sources across urbanization levels. Studies from the UK, validating different secondary data sources at varying levels of urbanization and across socio-economic levels, reported no notable differences across all study areas and fairly high agreement statistics, ranging from “moderate” to “excellent” [44,48]. In the US, even though studies have reported no marked differences across urban and rural areas, the magnitude of the validity scores has varied greatly between secondary data sources [38,46,50]. This suggests that the reason for the observed differences in terms of validity scores across urban and rural areas may be attributable to the data sources themselves, or to the geographic area of interest. For instance, since food retailers in rural areas are generally small and serve a limited number of local residents, some may choose not to be registered in commercial listings or other online secondary data directories [48]. Consequently, some (but not all) secondary data sources may be less able to correctly describe the food environment in a rural area.

Commercial parties’ methodological considerations for the classification of food retailers may also influence the validity and comparability of secondary data sources. To shed light on the usefulness of a commercial classification of food retailers for nutritional or public health research purposes, we also included agreement statistics on the “classification” level of analysis. While overall agreement on “classification” was relatively high, retailers such as animal product stores, convenience stores, and take away restaurants showed lower validity scores compared to other food retailer subcategories. These lower validity scores may be attributable to three aspects of classifying food retailers. Firstly, various retailers were combined into a food retailer category when constructing the field audit classification. For example, the food retailer category “animal product stores” included, among others, delicatessens. While we conceived delicatessens to be retailers selling mainly high-quality animal-based foods, such as cheese and salami, Locatus considered Italian and Polish shops also as delicatessens. Thus, a high number of false positives may have led to an underestimation of the PPV of a food retailer category. Secondly, misclassification may arise when retailers present multifaceted characteristics and thus they have no univocal definition. Studies from the UK, Canada, and the US [44,47,50] reported that convenience stores tend to have lower agreement statistics as compared to other food retailer subcategories. Convenience stores vary widely, and including (for instance) gas stations, pharmacies, and country stores. Additionally, in the Netherlands, convenience stores also offer a range of healthy and fresh products, unlike convenience stores in other countries. This variation may make them relatively difficult to recognize as such [29]. This is in contrast with the ease of classifying food retailers that present unique and clear characteristics (e.g., fast food chains) that may be easier to accurately detect. Thirdly, the multiple business features of some retailers (e.g., grocery stores that may offer the possibility to sit at a table for on-site consumption, or fast food restaurants that also offer take away service) may hinder the classification process [43], leading to some misclassification. In this case, reasoning in terms of what the main business purpose of a certain retailer is may facilitate the classification process.

Strengths of this study included its uniqueness, since to our knowledge this is the first study to assess the validity of commercially available data in the Netherlands. Next, the rigorous method used to separately calculate agreement statistics on “location”, “classification”, and both combined, have allowed light to be shed on the specific causes that may affect the validity of commercially available data. In addition, we examined a wide range of food retailers, offering an accurate listing of all common food retailers present in the study areas of interest. Lastly, during the field audit, the auditor was blinded to the commercial dataset in order to prevent the risk of being influenced in the data collection.

Nevertheless, some limitations need to be highlighted. First, it is worth noting that agreement statistics such as specificity and Kappa on the “location” level of analysis are affected by the low prevalence of food retailers and, in case of the urban areas, the very small number of true negatives (i.e., streets with no food retailers). Therefore, in these cases, the observed specificity and Kappa were excluded from further discussion. Second, due to research time constraints, a limited number of streets were purposively sampled in the eight neighborhoods that were characterized by being relatively small and easily accessible by public transport. However, given the coverage of both urban and rural areas, and the variety in number of food outlets per area, this selection is unlikely to have a major impact on the generalizability of the results. Third, since the field audits were conducted in February 2019 and the Locatus dataset was released in July 2018, some of the non-matching retailers may be attributed to the temporal mismatch between the two datasets. Retailers closing, opening, rebranding, and relocating during the seven-month time frame could have presumably increased the number of false positives and false negatives. Finally, retailers selling alcoholic beverages, establishments whose main business purpose was not selling food, and mobile vendors were not considered in the present study. Future validation studies should consider alternative sources of food and drinks in order to investigate whether they are correctly listed in secondary data sources.

Regardless of our findings, secondary data sources should always be used with caution and we would advise researchers to always validate their commercial data sources for use in health research. Notably, the use of field audits to validate secondary data sources has been described as being less suitable in large urban areas [34]. Collecting data via field audit observations in large geographic areas and in areas with high food retailer densities may be very labor intensive, and consequently not always feasible. Our study was characterized by a relatively small geographic area of interest and a limited number of streets, and thus no particular issues were encountered when conducting the field audit. If, however, secondary data sources cannot be complemented with extensive field work, alternative strategies such as combining at least two secondary data sources to improve the levels of accuracy [46], or the use of remote online-based techniques or street-viewing applications [52], should be considered in order to achieve an adequate alternative to field validations.

## 5. Conclusions

In this study, we assessed the validity of a secondary data source (“Locatus”) containing information on the geographical locations and types of food retailers against field audit data. In conclusion, overall agreement statistics across urban and rural areas ranged from “good” to “excellent” for all three levels of analysis (i.e., location, classification, and both combined). Therefore, policymakers and researchers should feel confident in using Locatus as a secondary source for assessing location and classification data of food retailers in the Netherlands. In addition, we highlighted a number of methodological considerations that may explain variation in the validity of secondary data sources, and that could be taken into account when comparing or harmonizing different data sources on food environments.

## Figures and Tables

**Table 1 ijerph-17-01946-t001:** Field audit-derived classification of food retailers based on Locatus’s definitions.

Retail Category	Field Audit Subcategories	Locatus’s Categories
Grocery stores	Supermarkets	Supermarkets
	Local product shops	Toko, foreign country shops (others)
	Fruit and vegetable stores	Vegetable/fruit stores
	Bakeries	Bakeries
	Animal product stores	Cheese stores, poultry stores, butcheries, delicatessen, fish stores
	Natural product stores	Health food stores, coffee/tea stores, nut stores
	Convenience stores	Minimarkets, night shops
	Confectionery stores	Pastry stores, chocolate stores, ice-cream saloons, candy stores
Food outlets	Restaurants	Full-service restaurants, café-restaurants, pancake restaurants, hotel-restaurants
	Fast food restaurants	Fast food restaurants, grillroom/shoarma/pita places
	Take away restaurants	Delivery/take away outlets
	Cafés	Coffee houses, lunchrooms

**Table 2 ijerph-17-01946-t002:** Measures of validity of commercially available data as compared to field audit data.

Field Audit Data					Validity Score
		Present	Absent	Sensitivity	$\;\frac{TP}{TP+FN}\;$
Commercially available data	Present	TP	FP	Specificity	$\;\frac{TN}{TN+FP}\;$
	Absent	FN	TN	PPV	$\;\frac{TP}{TP+FP}\;$
				Kappa	$\;\frac{po-pe}{1-pe}\;$
				Concordance	$\;\frac{TP}{TP+FP+FN}\;$

TP, true positive; FP, false positive; FN, false negative; TN, true negative; PPV, positive predictive value; p_o_, observed agreement = $\frac{TP+TN}{TP+FP+FN+TN}\;$; p_e_, expected agreement = $\left(\frac{TP+FP}{TP+FP+FN+TN}\times \frac{TP+FN}{TP+FP+FN+TN}\right)+\left(\frac{FN+TN}{TP+FP+FN+TN}\times \frac{FP+TN}{TP+FP+FN+TN}\right)$.

**Table 3 ijerph-17-01946-t003:** Descriptive statistics derived by comparing the Locatus data against the field audit data.

Category and Subcategory		No. of Food Retailers Listed in Locatus (Collected Until July 2018)	No. of Food Retailers Found in the Field (Collected between Feb 22 and March 2, 2019)	Matching *	Non-Matching				
					Error in Location *	Error in Classification *	Error in Both Location and Classification *	Not Found in the Field *	Found in the Field But Not Listed ^†^
	Total	322	315	246	1	42	0	33	26
Urbanization ^‡^	Urban	276	265	207	1	37	0	31	20
	Rural	46	50	39	0	5	0	2	6
Grocery stores (*N*)	Supermarkets	17	15	15	0	0	0	2	0
	Local product shops	0	6	0	0	0	0	0	1
	Fruit and vegetable stores	2	3	2	0	0	0	0	1
	Bakeries	10	7	7	0	1	0	2	0
	Animal product stores	22	18	13	1	8	0	0	3
	Natural product stores	8	10	7	0	1	0	0	1
	Convenience stores	6	3	3	0	3	0	0	0
	Confectionery stores	13	9	7	0	0	0	6	1
Food outlets (*N*)	Restaurants	137	152	121	0	3	0	13	10
	Fast food restaurants	33	30	27	0	4	0	2	0
	Take away restaurants	22	13	10	0	8	0	4	1
	Cafés	52	49	34	0	14	0	4	8

Note that not all numbers add up, as some food retailers found in the field but not listed by Locatus were actually listed by Locatus as another category of food outlet (e.g., of the six local product shops found in the field that were not listed by Locatus, only one was not listed at all by Locatus, while another five were listed by Locatus but not as local product shops—three were listed as convenience stores and two as animal product stores). * Frequency and percentage of food retailers listed in Locatus that matched or did not match the food retailers ascertained in the field. ^†^ Frequency and percentage of food retailers found in the field that were not listed in Locatus. ^‡^ Urbanization levels as defined by the Centraal Bureau voor de Statistiek (CBS).

**Table 4 ijerph-17-01946-t004:** Agreement statistics on “location” of food retailers for the Locatus dataset.

By Category	By Subcategory	TP *	FN	FP	TN	Locatus Dataset				
						Sensitivity	Specificity	PPV	Kappa	Concordance
**Overall**		288	27	33	61	0.914	0.649	0.897	0.576	0.827
**Urbanization**	Urban	244	21	31	3	0.921	0.088	0.887	0.010	0.824
	Rural	44	6	2	58	0.880	0.967	0.957	0.852	0.846
**Grocery stores**	Supermarkets	15	0	2	61	1.000	0.968	0.882	0.921	0.882
	Local product shops	0	1	0	61	-	1.000	-	-	-
	Fruit and vegetable stores	2	1	0	61	0.667	1.000	1.000	0.792	0.666
	Bakeries	8	0	2	61	1.000	0.968	0.800	0.873	0.800
	Animal product stores	21	4	0	61	0.840	1.000	1.000	0.882	0.840
	Natural product stores	8	1	0	61	0.889	1.000	1.000	0.933	0.888
	Convenience stores	6	0	0	61	1.000	1.000	1.000	1.000	1.000
	Confectionery stores	7	1	6	61	0.875	0.910	0.538	0.616	0.500
**Food outlets**	Restaurants	124	10	13	61	0.925	0.824	0.905	0.757	0.843
	Fast food restaurants	31	0	2	61	1.000	0.968	0.939	0.953	0.939
	Take away restaurants	18	1	4	61	0.947	0.938	0.818	0.839	0.782
	Cafés	48	8	4	61	0.857	0.938	0.923	0.800	0.800

TP, true positive; FN, false negative; FP, false positive; TN, true negative; PPV, positive predictive value. * Number of stores correctly located (246) and number of stores wrongly classified (42) but found to be in the correct location.

**Table 5 ijerph-17-01946-t005:** Agreement statistics on “classification” of food retailers for the Locatus dataset.

By Category	By Subcategory	TP *	FP	Locatus Dataset
				PPV
**Overall**		247	42	0.855
**Urbanization**	Urban	208	37	0.849
	Rural	39	5	0.886
**Grocery stores**	Supermarkets	15	0	1.000
	Local product shops	-	-	-
	Fruit and vegetable stores	2	0	1.000
	Bakeries	7	1	0.875
	Animal product stores	14	8	0.636
	Natural product stores	7	1	0.875
	Convenience stores	3	3	0.500
	Confectionery stores	7	0	1.000
**Food outlets**	Restaurants	121	3	0.976
	Fast food restaurants	27	4	0.871
	Take away restaurants	10	8	0.556
	Cafés	34	14	0.708

TP, true positive; FP, false positive; PPV, positive predictive value. * Number of stores correctly classified (246) and number of stores wrongly located (1) but correctly classified.

**Table 6 ijerph-17-01946-t006:** Agreement statistics on both the “location and classification” of food retailers for the Locatus dataset.

By Category	By Subcategory	TP *	FN	FP	Locatus Dataset		
					Sensitivity	PPV	Concordance
Overall		246	1	42	0.996	0.854	0.851
Urbanization	Urban	207	1	37	0.995	0.848	0.845
	Rural	39	0	5	1.000	0.886	0.886
Grocery stores	Supermarkets	15	0	0	1.000	1.000	1.000
	Local product shops	-	-	-	-	-	-
	Fruit and vegetable stores	2	0	0	1.000	1.000	1.000
	Bakeries	7	0	1	1.000	0.875	0.875
	Animal product stores	13	1	8	0.929	0.619	0.590
	Natural product stores	7	0	1	1.000	0.875	0.875
	Convenience stores	3	0	3	1.000	0.500	0.500
	Confectionery stores	7	0	0	1.000	1.000	1.000
Food outlets	Restaurants	121	0	3	1.000	0.976	0.976
	Fast food restaurants	27	0	4	1.000	0.871	0.871
	Take away restaurants	10	0	8	1.000	0.556	0.556
	Cafés	34	0	14	1.000	0.708	0.708

TP, true positive; FN, false negative; FP, false positive; PPV, positive predictive value. * Number of stores correctly located and classified (246).

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
