# Peer review of "Field Validation of Commercially Available Food Retailer Data in the Netherlands"

_ijerph, 2020, doi:10.3390/ijerph17061946_

Round 1

Reviewer 1 Report

Well done.

Reviewer 2 Report

Thank you for sending me this paper to review, I found it interesting to read and can see it provides important information for those using secondary data sources of food retailers. I have noted some minor comments below mostly to help with the flow and clarity of the paper. Please note I am not a statistician and so have not provided comments on the statistical methods or results. Overall, well done to the authors!

Page 2

Line 52: suggest you add ‘usually’ after ‘the research group and they usually have other purposes…’ (as some government repositories may have a health purpose).

For ease of understanding, I would move lines 54-57 re primary data collection before lines 52 -53 on secondary data collection. This then will flow onto the next para.

Page 3

For the information in Lines 90-95 this may be presented better in a table. Similarly you could also incorporate the information in Lines 95 and 96 into the table.

Line 120: what are systemic area scans? Can you explain or provide an example.

Page 4

The description of the field audit process was a bit confusing e.g. Line 129 you mention established were classified – according to what? You need to specify. You go on to say external clues but provide no explanation of what these are or even an example.

Line 130: step 3 you mention the auditor is to consult the menu – for what? You need to specify.

Line 131: to aid with clarity, perhaps the elements in step 4 should go first when describing the field audit process, in terms of the information you were seeking in the field audit ,and then you could go on to outline the steps to gain that information.

Line 132: could you please explain does the sign you refer to need to be a ‘closed’ sign or does to need to say the establishment is permanently closed?

Line 136: was this classification determined prior to the field audit? If yes, perhaps put it before the field audit process section? If not, state it was developed following the audit.

Page 5

Line 152: No need to change this but it’s interesting that you include tea and coffee in ‘natural product stores’, I would not have thought this.

Line 165: perhaps add you are not able to sit and consume a meal (if that is the case).

Line 168: again no need to change but interesting to note that cafes in other countries have a more diverse food offering than you mention here.

Table 3: It would be helpful to put the date the data was collected under the headings of ‘No. of food retailers listed in Locatus’ and ‘No. of food retailers found in the field’

? page

Line 227: Is it possible to provide information on what these were classified as? So the reader can understand was it a similar store eg café instead of restaurant (indicating a misclassification), or had it totally changed eg café to animal product store (indicating a total change of store).

Page 17

Line 342: you mention the mismatches in between the secondary data and the field data could be due to the time that passed between the two sources of data and that some shops may have closed or changed names - do you know how common this is within the time period you examined? Would you expect to see the level of change in stores that you found in the data you collected? How does it compare to the average in the Netherlands (or in other countries)?

Page 18

Line 370: I think the wording in this sentence could be a little clearer. Instead of ‘some of them may not consider the possibility to be registered’ you may want to substitute ‘some may choose not to be registered’

Line 395: add grocery stores that may have the possibility

Reviewer 3 Report

This is an interesting and useful study which assesses the validity of a commercial dataset containing information on the geographic location and classification of food retailers. The researchers compare the commercial dataset to a field audit, and calculate agreement statistics based on matching on location, classification and both. The study is especially useful to researchers wanting to study the food environment in the Netherlands but also contributes to a wider discussion on the use of secondary sources for retail food environment studies more generally. The manuscript is well written and easy to follow. I have some minor comments:

Line 86: is it 8 neighborhoods total?
Line 97: This sounds like it is convenience sampling. I suggest being more transparent, and commenting in the discussion if this type of sampling might have led to bias in your results. E.g. do you have a sense of selected area's socioeconomic level and whether your selection included both high and low socioeconomic level areas? Could the fact that you selected easy to access areas that were close to public transport influence results?
Line 120: What are 'systemic area scans'? I think it is important to provide more information about how Locatus collects their data and who does this.
Line 158-168: How do you classify a food outlet that is both fast food restaurant and take away?
Table 3: Local product shops, the numbers don't add up. If the numbers in the first three columns are correct then "found in the field but not listed" would be 6? If indeed an error, also correct line 232.
Table 3: In this table, the percentage in parenthesis is difficult to interpret.
Perhaps line percentages would be more informative.
Lines 241-246: For food shop categories with very few numbers for example Fruit and vegetable stores/convenience stores, how reliable are your agreement statistics? I suggest commenting in limitations.
